energy

numerical study, partial diesel particulate filter, deposition distribution, filtration efficiency

**Author for correspondence:**
Yangbo Deng
e-mail: dengyb2020@163.com

# Two-dimensional numerical studies of particle motion and deposition in the channel of diesel particulate filters

Xiaolong Wang[1], Yangbo Deng[2] and Yang Liu[1]

[1]Marine Engineering College, and [2]Naval Architecture and Ocean Engineering College, Dalian Maritime University, Dalian 116026, Liaoning, People's Republic of China

XW, 0000-0003-4608-197X; YL, 0000-0001-9910-8430

A numerical investigation on the soot laden flow of gas in a partial diesel particulate filter (PDPF) is presented based on solving the momentum equations for a continuous phase in the Euler frame and the motion equations for the dispersed phase in the Lagrangian frame. The interaction between the gas phase and the particles is considered as a one-way coupling for dilute particle concentration, while the interaction between particles and porous wall is implemented through user-definedsubroutines. To accurately track motion of nanoscale particles, the Brownian excitation and drag force as well as partial slip are taken into account in the particulate motion equation. Two methods are used to verify the gas flow model and reasonable agreements for both comparisons are observed. The effects of inlet velocity, wall permeability and particle size on the filtration efficiency and deposition distribution of the particles along with wall surface of inlet channel are quantitatively studied. The results show that (i) wall permeability plays the primary role in determining the filtration efficiency of PDPF, (ii) both upstream velocity and particle size have an effect on the initial deposition position of particles and (iii) filtration efficiency of PDPF is not markedly proportional to gas flow into inlet channels at a low wall permeability.

## 1. Introduction

As one of the most harmful components of exhaust gas from diesel engines during the last decades, particulate matter (PM) emission has aroused wide concern due to its potential effects on the atmospheric environment and human health [1–4]. As a result, increasingly stringent regulations have been issued to restrict particulate emission by many countries, driving manufacturers to develop advanced after-treatment techniques.

**Figure 1.** Structure of DPF and PDPF. (*a*) Macro-structure of regular wall-flow DPF [6]. (*b*) The cell structure of regular wall-flow DPF. (*c*) Macro-structure of PDPF investigated in the current paper. (*d*) Cell structure of PDPF investigated in the current paper.

Based on years of practice, wall-flow diesel particulate filters (DPFs) have been considered as the most effective device to reduce PM emission from diesel engines to a required scope [5]. Figure 1*a,b* shows a regular wall-flow DPF composed of hundreds of alternately congested inlet and outlet channels. When exhaust gas from diesel engines flows through the porous wall from the inlet channel to the outlet channel, soot particles will be trapped inside or on the porous wall. However, as an incidental result of this structure and working mode, the exhaust back pressure will be increased inevitably with the continuous deposition of particles, worsening the power performance and economy of diesel engines. Accordingly, measures are needed to remove the deposited particles periodically. The process is known as the regeneration of DPF [7,8].

Optimum structure of DPF requires low-pressure drop and regeneration frequency. The best strategy to lower both pressure drop and regeneration frequency is to operate a continuous filtration/regeneration process of catalyst-coated DPFs [9]. Scholars have done a lot of exploratory research on this topic, such as reducing CPSI (cells per square inch) [10] and altering channel shape [11], using unequal width for inlet and outlet channels [12–14]. This study presents a DPF operated using a method of partial filtration named partial diesel particulate filter (PDPF). Figure 1*c,d* show the structure of PDPF studied in this paper. Different from the regular DPF, all the frontal plugs of outlet channels are removed and only the plugs of inlet channels are retained. For the presented PDPF, the total trap efficiency is smaller than that of popular DPF because part of exhaust gas flows through outlet channels. However, the PDPF may be attractive for its smaller back pressure [6].

In this work, a systematic numerical simulation on the gas and solid phase flow in a single-channel PDPF is carried out. First, the gas flow field inside a single-channel PDPF is studied. Next, particles are uniformly released into the computational zone from the inlet and are tracked until deposited on the porous wall surface of the inlet channel or escaped from the outlet channel. Finally, the influence of wall permeability, upstream velocity and particle size on filtration efficiency and particle deposition are investigated.

## 2. Mathematical model and computational methodology

To simplify computation, the following assumptions are introduced:

(i) The gas flow in the channel is laminar. This assumption is reasonable since the Reynolds number is less than 2000 [15].

(ii) The force of particles on the gas phase and collisions between particles is neglected due to the extremely small volume fraction of PM in exhaust gas [16].

(iii) The particles are trapped once they hit the surface wall porous media in the inlet channel. This assumption can be supported by the rigid-body impact models developed by Dunn [17] and Dahneke [18] demonstrated that particles with a normal velocity lower than $2\,\mathrm{m\,s^{-1}}$ when hitting the porous wall are all captured by the adhesive forces and adsorption on the ceramic surface.

(iv) The flow-through mechanism which contributes to the overall filtration efficiency of PDPF is not considered as has been done in [6].

## 2.1. Flow field modelling of the continuous phase

The continuity equation and momentum equation are given as follows

$$\frac{\partial u_i}{\partial x_i} = 0 \tag{2.1}$$

and

$$\rho_g u_j \frac{\partial u_i}{\partial x_j} = \frac{\partial P}{\partial x_i} + \mu \frac{\partial^2 u_i}{\partial x_j^2}, \tag{2.2}$$

where $\rho_g$ is gas mixture density, $\mu$ is the gas viscosity and $P$ denotes the gas pressure.

When gas mixture flows inside porous wall, an additional resistance needs to be considered, including the viscous and the inertial effect:

$$S_i = \frac{\mu}{k_w} u_{wi} + \frac{1}{2} \beta \rho_g |u_{wi}| u_{wi}, \tag{2.3}$$

where $k_w$ is the wall permeability, $u_w$ is the wall velocity and $\beta$ is the Forchheimer factor. The second term on the right-hand side of equation (2.3) represents the Forchheimer term which can be neglected for most engine operating conditions [19].

## 2.2. Modelling of forces on the discrete phase

The force balance equation for each particle is given as follows

$$\frac{dv_i}{dt} = F_{D_i} + F_{B_i}, \tag{2.4}$$

where $v_i$ is the particle velocity. $F_{D_i}$ is the drag force and written as [20]

$$F_{D_i} = \frac{3\mu C_D Re_p}{2d_p^2(2\rho_p + \rho_g)} \cdot (u_i - v_i) \cdot \frac{1}{SCF}, \tag{2.5}$$

where $d_p$ is the particle diameter, $\rho_p$ is the particle density and $C_D$ is drag coefficient and is given as follows [21]

$$\begin{cases} C_D = \dfrac{24}{Re_p} & (Re < 1) \\[3mm] C_D = \dfrac{24}{Re_p(1 + (1/6)Re_p^{2/3})} & (1 < Re < 1000) \end{cases} \tag{2.6}$$

The particle Reynolds number $Re_p$ is defined as follows

$$Re_p = \frac{\rho_g d_p |u_i - v_i|}{\mu}. \tag{2.7}$$

The Stokes–Cunningham slip factor $SCF$ is given as follows

$$SCF = 1 + \frac{2\lambda}{d_p}(1.257 + 0.4e^{-(11d_p/20\lambda)}), \tag{2.8}$$

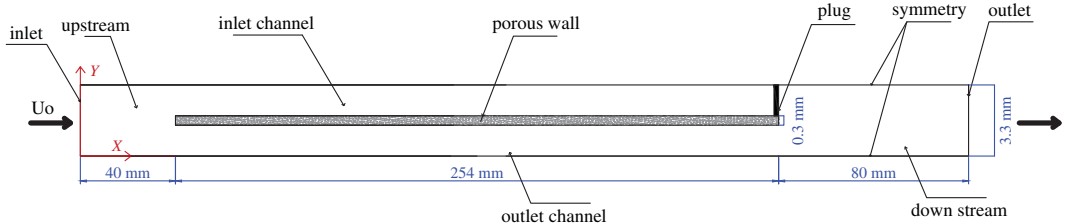

**Figure 2.** Computational domain and boundary conditions.

**Table 1.** Geometrical details of the computational domain.

| geometrical parameters | value |
| --- | --- |
| upstream length (mm) | 40 |
| downstream length (mm) | 80 |
| channel length (mm) | 254 |
| channel width (mm) | 1.5 |
| channel wall thickness (mm) | 0.3 |

where $\lambda$ is the molecular mean free path of the gas, $F_{B_i}$ is the Brown force and is given as follows [22]

$$F_{B_i} = \xi \sqrt{\frac{6\pi d_P \mu k_B T}{Vt}},$$ (2.9)

where $k_B$ is the Boltzmann constant, and $\Delta t$ is the time step magnitude. $\zeta$ is a Gaussian random number.

## 2.3. Flow boundary conditions

In order to reproduce faithfully the working conditions of the filter, appropriate boundary conditions are applied to domain boundaries. An adjacent inlet and outlet channel is considered as the computational zone in view of the symmetric structure of PDPF. To eliminate the boundary effect, a zone is added upstream of the flow entrance and downstream of the flow exit of PDPF. Figure 2 shows the computational zone and boundary conditions. The geometrical details are shown in table 1. As shown in figure 2, an inlet condition is placed in the far upstream boundary, normal to x-axis. It provides a velocity, uniform along with y directions and parallel to x-axis. Pressure-outlet boundary condition assumed to be atmospheric pressure is applied to the outlet condition. The upper and lower boundaries of the domain are set with symmetric boundary conditions. Finally, a no-slip and no-cross boundary condition is applied to the plug.

## 2.4. Computational methodology

The computational domain is discretized in 299 200 square cells, among which the minimum cell size is 0.015 mm in porous wall and 0.025 mm in channels, and the grid independence of the results is verified. Tables 2 and 3 show the operation conditions and physical parameters.

The software Fluent [23] is used to study the gas mixture flow and particle motion, while the interaction between the particles and the wall is implemented through user-definedsubroutines. SIMPLE is used to deal with the coupling between the pressure and gas velocity. The number of particles trapped on the surface of porous wall is counted and recorded through user-defined-memories.

# 3. Results and discussions

## 3.1. Model validation

Both the fragility and hardness of material and the stricture of channels make it very difficult to obtain the distribution of gas flow filed in a DPF by experimental measurement. Thus, we make an indirect flow

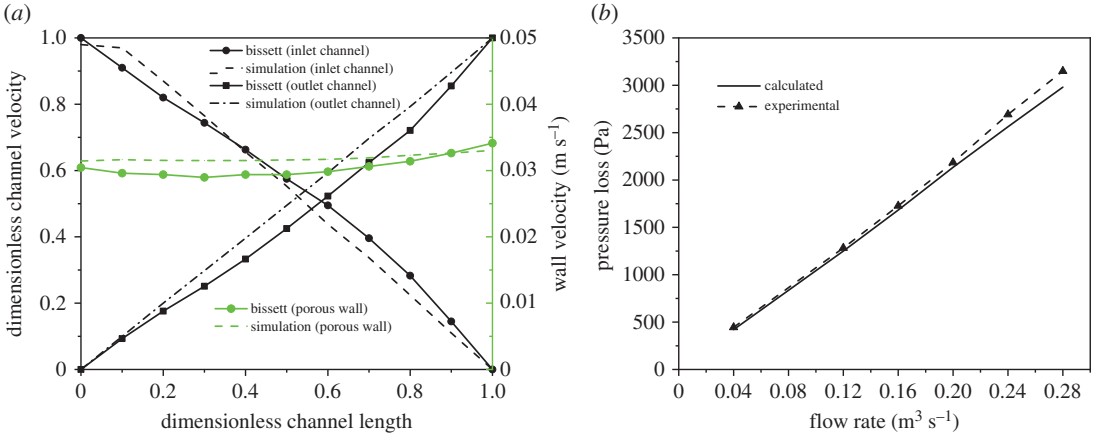

**Figure 3.** Predicted velocity filed, pressure loss and the results by experiment. (*a*) Velocity field comparison between two-dimensional and one-dimensional models. (*b*) Pressure loss comparison between simulating results and experiments [20].

**Table 2.** Operation conditions and physical parameters.

| operation conditions/physical parameters | value |
|---|---|
| upstream velocity (m s$^{-1}$) | 1, 2, 8 |
| gas temperature (K) | 600 |
| gas viscosity (kg m s$^{-1}$) | $3.2 \times 10^{-5}$ |
| gas density (kg m$^{-3}$) | 0.54 |
| wall permeability (m$^2$) | $1.25 \times 10^{-13}$, $5.0 \times 10^{-13}$, $5.0 \times 10^{-12}$ |
| particle diameter (nm) | 20, 100, 1000 |
| particle density (kg m$^{-3}$) | 2000 |

**Table 3.** Geometrical parameters.

| geometrical parameters | value |
|---|---|
| upstream length (mm) | 61 |
| downstream length (mm) | 305 |
| channel length (mm) | 305 |
| channel width (mm) | 2 |
| channel wall thickness (mm) | 0.43 |

field validation by using the one-dimensional single-channel flow model proposed by Bissett [24] and the characteristic curve by which we mean here the pressure drop between the two sides of the filter as a function of the overall flow rate of the filter, experimentally obtained by Liu *et al*. [16]. Figure 3*a* shows the comparison of the calculated velocity fields with that obtained from the Bissett's model. Local velocity of the two-dimensional model is divided by the maximum axial velocity. One can see that velocity fields predicted by the two models match well. In the experiments carried out by Liu *et al*., a six-cylinder, four-stroke cycle diesel engine was used to produce real exhaust. The pressure loss across the DPF was measured by piezometer rings mounted at the flow entrance and the flow exit of the DPF. By adjusting the opening of the by-pass valve, the flow rate of the exhaust gas across the DPF was changed. Figure 3*b* shows the comparison of the calculated pressure loss with that measured in the experiment. It is observed that the overall trends of both are consistent. However, the predicted values deviate slightly from the experiments as the flow rates get higher. This is possibly caused by model simplification and the increased non-uniformity of upstream velocity in these cases.

## 3.2. Particle motion trajectories

In this section, an inlet gas mixture velocity of $4\,\mathrm{m\,s^{-1}}$, flow rate of $10^{-8}\,\mathrm{kg\,s^{-1}}$ for the particle with a diameter of 100 nm is chosen in the simulation. For the convenience of subsequent analysis on the particle motion trajectory, the definition of displacement direction deflection angle is introduced, which is defined as,

$$\theta = \arctan\frac{y_0 - y_1}{x_1 - x_0},\tag{3.1}$$

where $y_0$ stands for the $y$ coordinate of an initial position of the particle and $y_1$ stands for the $y$ coordinate of the current position. Similarly, $x_0$ stands for the $x$ coordinate of initial position and $x_1$ is that of the current position of the particle.

The displacement direction deflection angle for each particle at each tracking time is calculated and stored as a scalar through user-definedsubroutines. By using $\theta$ as state variable, figure 4 illustrates the motion trajectories of 15 particles which are evenly distributed when released near the flow entrance of PDPF under various permeabilities. It is easily seen that all particles move in a parallel straight line ($\theta = 0$) in the front of upstream region in all cases. However, at the flow entrance, particles are diverted to different extents. The particles near the horizontal centre line of channel wall always deviate more. At the high permeability ($k_\mathrm{w} = 5.0 \times 10^{-12}\,\mathrm{m^2}$), the number of particles into the inlet channel is almost the same as the outlet channel. But at the low permeability ($k_\mathrm{w} = 1.25 \times 10^{-13}\,\mathrm{m^2}$), most of the particles go into the outlet channel and even for those close to the upper boundary, this will lead to a low filtration efficiency, as will be discussed in the next section.

## 3.3. Effect of wall permeability on the performances of partial diesel particulate filter

The wall permeabilities determine the difficulty for gas to flow inside the wall and surely have an effect on the flow field, which further affects the particle motion trajectory. In order to explore the effect of wall permeability on filtration efficiency and deposition distribution, four different permeabilities of $1.25 \times 10^{-13}$, $5.0 \times 10^{-13}$, $1.25 \times 10^{-12}$ and $5.0 \times 10^{-12}\,\mathrm{m^2}$ are chosen with an inlet velocity of $4\,\mathrm{m\,s^{-1}}$ and particle diameter of 100 nm.

Figure 5a shows the filtration efficiency of PDPF with permeability. It is clear that the filtration efficiency reaches the maximum when the wall permeability is $5.0 \times 10^{-12}\,\mathrm{m^2}$. However, as the permeability decreases, the filtration efficiency shows a marked downward trend. This is quite the opposite in a regular DPF where the filtration efficiency increases gradually during both deep trapping stage and cake trapping stage along with a decrease in permeability. Factually, to a large extent, the filtration of the PDPF relies on the redistribution of the flow rate at the flow entrance [6]. When the wall permeability is low, less soot laden gas will flow into the inlet channel. As a better illustration, figure 6 shows the velocity field of continuous phase under two different wall permeabilities (the computational domain is scaled along with the transverse direction using a scaling factor equal to 8). Figure 5b shows the distribution of particle deposition along with the wall surface of the inlet channel. X-axis stands for dimensionless position of channel length and $y$-axis represents dimensionless number concentration percentage of particles. It can be seen obviously that most particles are deposited in the rear of the channel at a high permeability and deposited towards the front of the inlet channel as the permeability decreases. It meets our common knowledge that a lower permeability leads to an increased resistance of gas flow in the inlet channel, and then shortens the moving distance of particles. Subject to large flow resistance in this case [6], particles have difficulty reaching the end of the inlet channel.

## 3.4. Effect of particle size on the performance of partial diesel particulate filter

The movement trial of a particle is affected by its own size due to the inertia. Considering that most of the particles in diesel engine exhaust are nanoscale or submicron, we choose three sizes (20, 100 and 1000 nm) of particles to be tracked under the same flow field conditions. Since the permeability has significant influence on the flow field, a higher permeability of $5.0 \times 10^{-12}\,\mathrm{m^2}$ and lower permeability of $1.25 \times 10^{-13}\,\mathrm{m^2}$ are used to evaluate the influence of particle size on the filtration efficiency and deposition distribution.

Figure 7 presents the filtration efficiency of PDPF and deposition distribution of particles with various sizes. The difference in filtration efficiency of PDPF for the three cases is not significant under either high permeability or low permeability. As far as deposition distribution is concerned, under the high

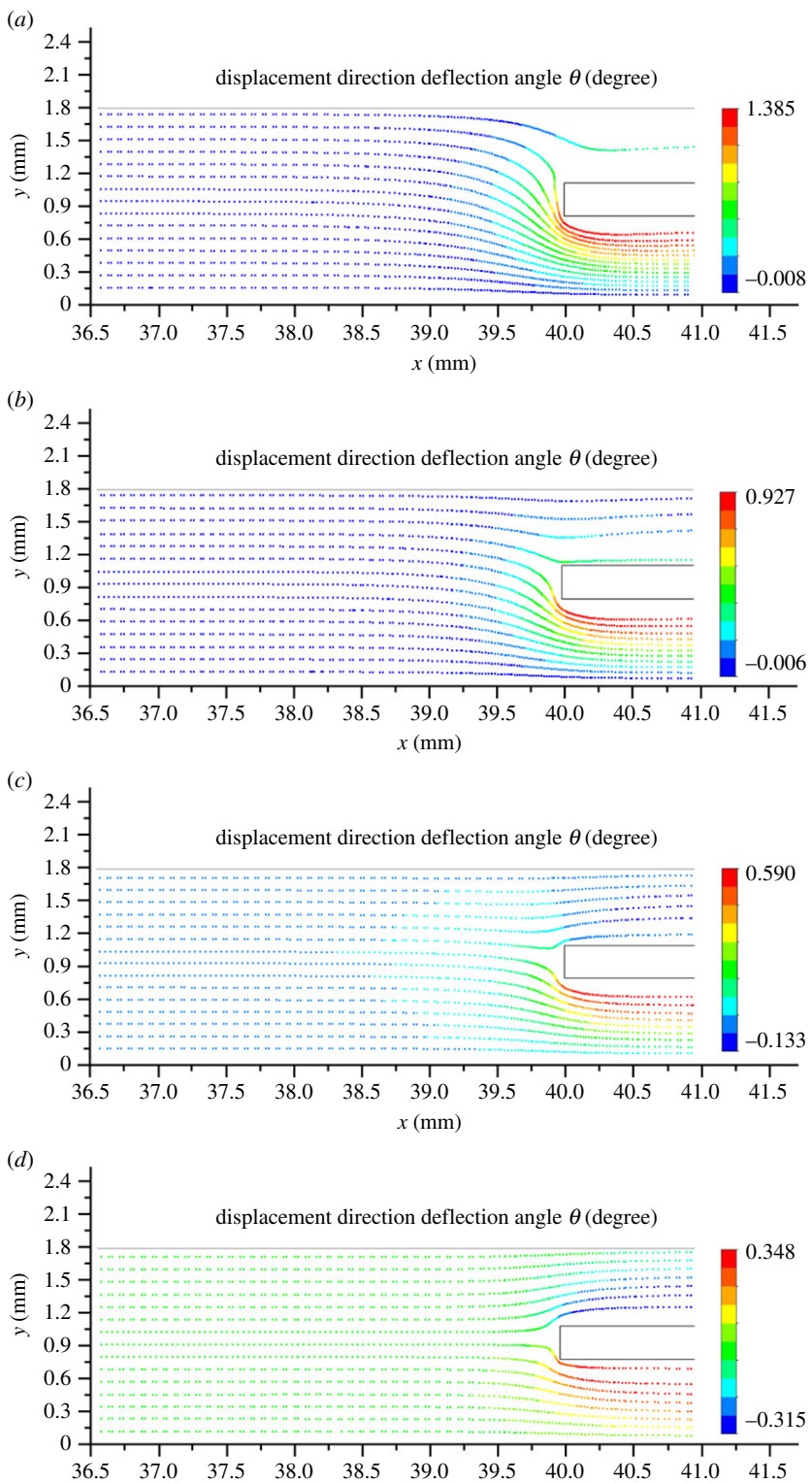

**Figure 4.** Trajectory tracking of particles at the flow entrance of PDPF under various permeabilities. (a) Trajectory tracking of particles under $k_w = 1.25 \times 10^{-13}$ m$^2$. (b) Trajectory tracking of particles under $k_w = 5.0 \times 10^{-13}$ m$^2$. (c) Trajectory tracking of particles under $k_w = 1.25 \times 10^{-12}$ m$^2$. (d) Trajectory tracking of particles under $k_w = 5.0 \times 10^{-12}$ m$^2$.

permeability ($k_w = 5.0 \times 10^{-12}$ m$^2$), there is a marked difference in the initial deposition position for particles of different sizes. As shown, particles with a diameter of 20 nm start to deposit at $x = 0.05$ while particles with a diameter of 1000 nm initially deposit until $x = 0.18$. This can be due to the fact that small particles have small inertia and strong Brownian motion. However, the same phenomena have not been observed in the

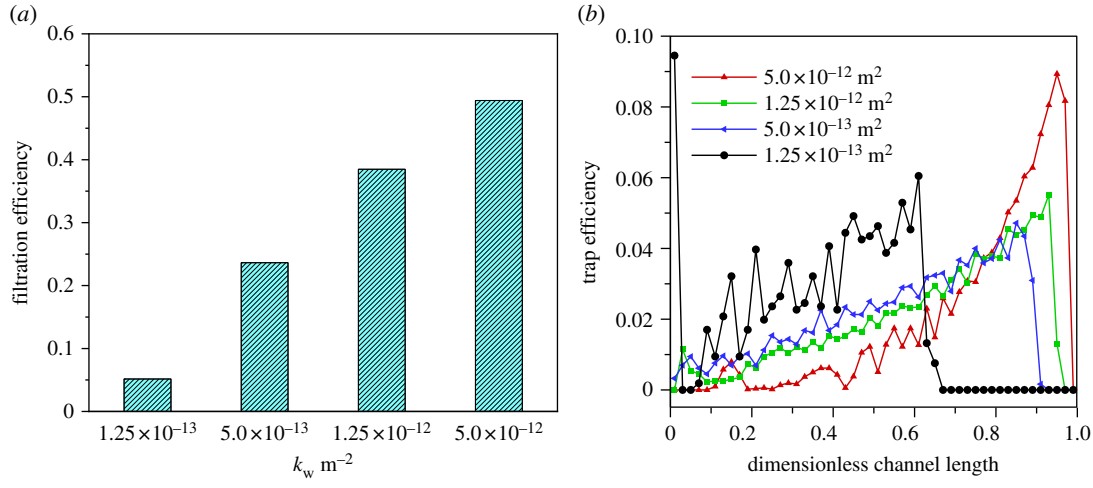

**Figure 5.** Filtration efficiency and deposition number concentration percentages along with the inlet channel under various wall permeabilities. (*a*) Filtration efficiency. (*b*) Deposition number concentration percentages.

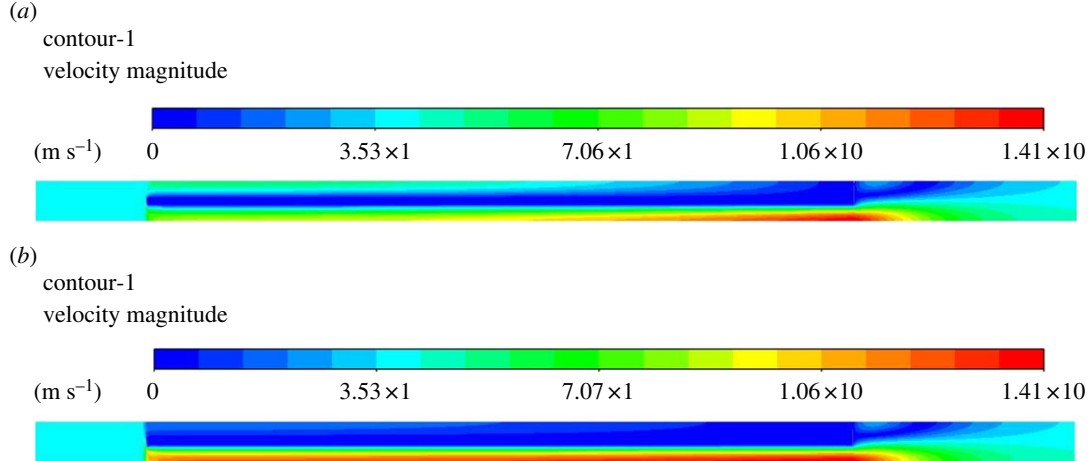

**Figure 6.** Velocity field of continuous phase under two different wall permeabilities at $U_0 = 4$ m s$^{-1}$. (*a*) Velocity field under $k_w = 1.25 \times 10^{-12}$ m$^2$. (*b*) Velocity field under $k_w = 2.0 \times 10^{-13}$ m$^2$.

case of $k_w = 1.25 \times 10^{-13}$ m$^2$, indicating that the effect of particle inertia and Brownian motion on the deposition distribution along with the inlet channel are weakened at low permeability.

## 3.5. Effect of inlet velocity on the performance of partial diesel particulate filter

The trajectory of the particle is determined by the flow field and therefore certainly affected by the upstream velocity. In this section, upstream velocities of 1 m s$^{-1}$, 2 m s$^{-1}$ and 8 m s$^{-1}$ are chosen to carry out the study on the influence of inlet velocity on filtration efficiency and deposition distribution at high and low wall permeability, respectively. As shown in figure 8*a*,*b*, the filtration efficiency increases slightly with the increase of upstream velocity under both high permeability and low permeability. This makes sense because particles with high velocity have more energy to get out of the streamline of the fluid into the inlet channel at the flow entrance. Figure 8*c*,*d* illustrate the distribution of particle depositions along with the wall surface of inlet channel. For $k_w = 5.0 \times 10^{-12}$ m$^2$, a noticeable difference in the position where particles initially deposit is observed. When upstream velocity is 8 m s$^{-1}$, no particle is trapped until $x = 0.2$, much more backward than that under the upstream velocity of 1 m s$^{-1}$. In the case of $k_w = 2.0 \times 10^{-13}$ m$^2$, a more concentrated and compact deposition distribution profile is observed and almost all particles are deposited in the region of $x < 0.8$ under various upstream velocities. However, no significant difference has been found from the overall

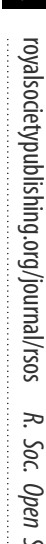

**Figure 7.** Filtration efficiency for various sizes of particle and deposition number concentration percentages along with the inlet channel under a high permeability and a low permeability. (*a*) Filtration efficiency for $k_{\rm w} = 5.0 \times 10^{-12}$ m$^2$. (*b*) Filtration efficiency for $k_{\rm w} = 2.0 \times 10^{-13}$ m$^2$. (*c*) Deposition number concentration percentages for $k_{\rm w} = 5.0 \times 10^{-12}$ m$^2$. (*d*) Deposition number concentration percentages for $k_{\rm w} = 2.0 \times 10^{-13}$ m$^2$.

distribution trend in these cases, indicating that deposition distribution of the particles is not sensitive to changes in upstream velocity at low permeability.

## 3.6. Comprehensive evaluation on the performance of partial diesel particulate filter

In the case of the upstream velocity being 4 m s$^{-1}$, the permeability being $5.0 \times 10^{-13}$ m$^2$ and the diameter of particle released being 100 nm, the filtration efficiency $\eta_{\rm f}$, the pressure drop reduction efficiency $\eta_{\rm P}$ defined as in equation (3.2) and the share of gas flow into the inlet channel $\eta_{\rm s}$ defined as in equation (3.3) under various permeabilities are shown in figure 9.

$$\eta_{\rm P} = \frac{\Delta P_{\rm re} - \Delta P_{\rm un}}{\Delta P_{\rm re}} \tag{3.2}$$

and

$$\eta_{\rm s} = \frac{f_{\rm inlet}}{f_{\rm inlet} + f_{\rm outlet}}. \tag{3.3}$$

In equation (3.2), $\Delta P_{\rm un}$ stands for the pressure drop through a PDPF and $\Delta P_{\rm re}$ stands for the pressure drop through a regular DPF. In equation (3.3), $f_{\rm inlet}$ stands for the flow rate in inlet channel and $f_{\rm outlet}$ is the flow rate in outlet channel.

It can be found in figure 9 that higher filtration efficiency always comes before the higher pressure drop. The relationship between the filtration efficiency and the pressure drop reduction efficiency is

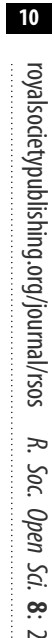

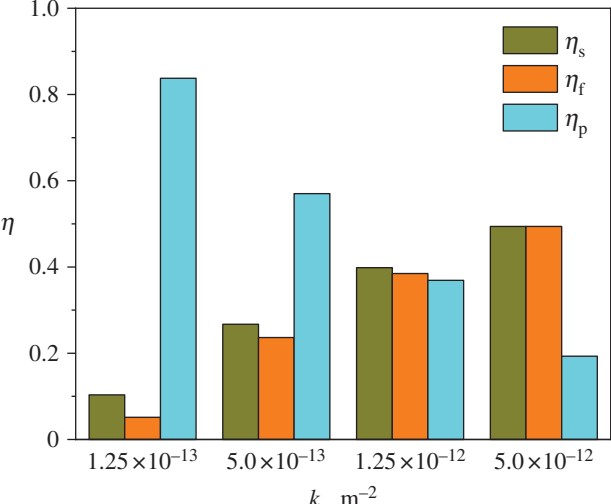

**Figure 8.** Filtration efficiency for various upstream velocities and deposition number concentration percentages along with the inlet channel under a high permeability and a low permeability. (*a*) Filtration efficiency for $k_w = 5.0 \times 10^{-12}$ m². (*b*) Filtration efficiency for $k_w = 2.0 \times 10^{-13}$ m². (*c*) Deposition number concentration percentages for $k_w = 5.0 \times 10^{-12}$ m². (*d*) Deposition number concentration percentages for $k_w = 2.0 \times 10^{-13}$ m².

**Figure 9.** Comprehensive analysis of filtration efficiency in relation to flow distribution and pressure loss under various wall permeabilities.

more like one of 'as one falls, another rises', in which the permeability plays a key role to coordinate between them. From further observation, we notice another interesting result that at high permeability ($k_w = 5.0 \times 10^{-13}$ m²), the filtration efficiency is nearly equal to the share of gas flow into the inlet

channel, indicating that the particles closely follow the motion of the gas. However, this 'harmony' is gradually broken as permeability decreases. At $k_w = 1.25 \times 10^{-13}$ m$^2$, the filtration efficiency is almost only half as much as the share of gas flow in the inlet channel, implying that the separation of particle motion from gas stream becomes intense at low permeabilities. This makes us aware that a too-low permeability will worsen the comprehensive working performance of PDPFs. These results have a guiding significance for further structural design optimization of PDPFs.

# 4. Conclusion

A two-dimensional gas-particle two-phase flow model is applied to study the performance of PDPF. Steady CFD-based simulations of particle motion in Lagrangian frame are run to investigate the effects of several operating conditions such as wall permeability, upstream velocity and particle size on filtration efficiency and deposition distribution of particles in a single-channel PDPF. From numerical results, the conclusions are as follows,

(i) The wall permeability has a most primary effect on the filtration efficiency of PDPF by redistributing the flow rate at the flow entrance. The dependence of the particle motion trajectory on upstream velocity is significant at a high wall permeability. However, this effect is alleviated in the cases of small permeabilities. The effect of particle size on its motion trajectory is attributed to inertia and Brownian motion. For small particles, the initial deposition position advances toward the front of the inlet channel.

(ii) The main function of a DPF is to filter particles. In the PDPF, most of the airflow and particles will flow through the outlet channel since the inlet channel is blocked, which is particularly notable at low wall permeabilities, resulting in the trap efficiency being smaller than that of the popular DPF. However, the attractive point of this PDPF is its smaller back pressure. This contradictory relationship between the trap efficiency and the back pressure can be harmonized by its wall permeability.

(iii) The filtration efficiency of PDPF is not proportional to the gas flow into inlet channel at low wall permeabilities, implying that the separation of particle motion from gas streamline becomes intense in these cases.

(iv) Under the conditions investigated in this work, with the progress of particle deposition, the wall permeability decreases, which will be bound to reduce the trap efficiency of PDPF. Accordingly, the regeneration procedure must be started when the particle deposition reaches a certain level. One promising way to prolong the working cycle of PDPF may be to optimize its structure. Future work will attempt to investigate the performance of PDPF with asymmetric channels.

Data accessibility. All data and code are available from the Dryad Digital Repository: https://doi.org/10.5061/dryad.pnvx0k6n4.

Authors' contributions. X.W. and Y.D. set up the model; Y.L. wrote the manuscript. All authors gave final approval for publication.

Competing interests. We declare we have no competing interests.

Funding. This work is supported by Dalian Science and Technology Innovation Fund (grant no. 2020JJ26SN065).

Acknowledgement. The author is grateful to many colleagues with whom he had the privilege to interact and collaborate over the years and whose work is partially referenced in this article.

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
