## [Peer Review File · Royal Society Open Science]

Review History

RSOS-211162.R0 (Original submission)

Review form: Reviewer 1

Is the manuscript scientifically sound in its present form?

No

Are the interpretations and conclusions justified by the results?

No

Is the language acceptable?

No

Do you have any ethical concerns with this paper?

No

Have you any concerns about statistical analyses in this paper?

No

Recommendation?

Major revision is needed (please make suggestions in comments)

Comments to the Author(s)

Review Comments: The work represents a 2D study of a PDPF, however, a few points need to be clarified before its consideration for publications.

Comment 1: The main function of a DPF is to filter particles. In this PDPF, most of the airflow and particles will flow through the outlet channel since the inlet channel is blocked. The trap efficiency should be much smaller than the popular DPF. The possible attractive point of this PDPF is its smaller back pressure. However, this is not mentioned in the conclusion. The authors need to provide a comparison of the trap efficiency and back pressure between PDPF and DPF and highlight significances and innovations of the PDPF.

Comment 2: The contents and some figures of this manuscript show high similarity with the reference 20 published in 2009. The authors need to rewrite and restructure most of their contents from Introduction to Section 3.1.

Comment 3: The structure of figure captions is not in general format. The authors need to modify that.

Comment 4: Figure 2 is hard to understand, the authors need to indicate the airflow directions, the parameters mentioned in table 1. Please follow the example from the reference 20 Figure 1(c), that is how a good model structure looks like. Based on the setups of the boundary conditions, the boundary of the inlet channel and the outlet channel (0.04 m ~ 0.294 m) should be wall, not symmetry. The authors need to clarify this.

Comment 5: Line 71, the equations of boundary conditions are unclear, the authors need to clarify what is u_1 and what is u_2 . There is an overuse of brackets. It is suggested to list the boundary conditions of one case in the same bracket. The equation number should be located in the vertical middle.

Comment 6: The figures of the filtration efficiency are lack of the unit. The authors need to clarify if the unit is in % or not. The quality of the Figure 5,6,7 needs to be improved. It is hard to recognize the bar charts, it is suggested to use the line charts for different wall permeabilities.

Comment 7: The authors need to carefully use the signal “-” in table 2. There are only 3 different particle sizes (20 100 1000) and 3 different upstream velocities (1 2 8). There are no enough variables to support the analysis.

Comment 8: The data in Figure 6(d) and Figure 7(d) are missed from 0.8 to 1.0. The authors need to explain why the data are missed.

Comment 9: The fraction efficiencies of different velocities, different particle sizes are almost the same. The impact of an open outlet channel has significant impact on the local velocity field in this PDPF. The authors are suggested to present the velocity field (velocity contours in ANSYS Fluent) for a better understanding.

Comment 10: ANSYS Fluent is a commercial software, the authors need to cite it as required by the ANSYS company.

Review form: Reviewer 2

Is the manuscript scientifically sound in its present form?

Yes

Are the interpretations and conclusions justified by the results?

Yes

Is the language acceptable?

Yes

Do you have any ethical concerns with this paper?

No

Have you any concerns about statistical analyses in this paper?

No

Recommendation?

Accept with minor revision (please list in comments)

Comments to the Author(s)

The work concerns the modelling and simulations on the soot laden flow of gas in a partial diesel particulate filter. The research covered an important area and the article may be of interest to many readers. The manuscript has been correctly-structured. Accurate conclusions are presented, which are supported by research results which are original achievement of the Authors. This paper is well written and it is worth to be published in the journal "Royal Society Open Science" with the very minor modifications needed and indicated below:

1. The abstract is well-written, however the main conclusion in the abstract is too long and should be simplified.
2. Necessary references need to be marked, such as the references on Figure 1a and 1b(lines 31-32), and the assumption (2)(lines 31-32).
3. The meaning of each symbol in the formula should be indicated, such as P and μ in the equation (2) ; k_w (permeability ?) in the equation (5) ; Re_p in the equation (10) ; dp in the equation (11) ; λ in the equation (12) ; k_B and Δt in the equation (13) , etc.
4. The description of "The continuity equation" in line 70 should be changed to "The continuity equation and momentum equation"
5. Some formulas are marked with incorrect symbol and need to be confirmed by the author. For example: ρ in formula (9) should be ρg ? Re in formula (10) should be Re_p ? SCF in line 86 should be italic?
6. Some grammar errors need to be checked throughout, such as "A" in line 54 should be "An"; "has" in line 102 should be "have"; "was" in line 209 should be "is", etc.
7. The url should be added in the "()" in line 241.
8. Is the unit of the flow rate(kg/m³) correct?

Review form: Reviewer 3

Is the manuscript scientifically sound in its present form?

Yes

Are the interpretations and conclusions justified by the results?

Yes

Is the language acceptable?

Yes

Do you have any ethical concerns with this paper?

No

Have you any concerns about statistical analyses in this paper?

No

Recommendation?

Major revision is needed (please make suggestions in comments)

Comments to the Author(s)

This is an interesting paper addressing an important issue. It can be considered for publication in Royal Society Open Science. However, I have the following comments that the authors should carefully implement in the revised manuscript before publication.

1) Introduction - The sentences "Accordingly, measures are needed to remove the deposited particles periodically. The process is known as the regeneration of DPF." should be substantiated by also citing the following work: Chemical Engineering Science, Volume 137, 2015, Pages 69-78.

2) Introduction - The authors wrote "Optimum structure of DPF requires low pressure drop and regeneration frequency.". This is surely true. However, in order to give a more complete picture, they should also highlight in the revised Introduction that, as recently reviewed (see, Catalysts, 2020, 10(11), Article number: 1307), the best strategy to lower both pressure drop and regeneration frequency is to operate a continuous filtration/regeneration process of catalyst-coated DPFs.

3) Computational methodology and conditions - Are the computational results grid-independent? The authors should comment on this key issue.

4) Results and discussions/Conclusions - In the discussion, the practical impact of the results obtained in this work should be better pointed out. This should also be done in the section "Conclusions".

5) Conclusions - The authors should also give an outlook on future research work.

I'm willing to review the revised manuscript.

Decision letter (RSOS-211162.R0)

Dear Dr Wang

The Editors assigned to your paper RSOS-211162 "2D numerical studies of particle motion and deposition in the channel of diesel particulate filters" have now received comments from reviewers and would like you to revise the paper in accordance with the reviewer comments and any comments from the Editors. Please note this decision does not guarantee eventual acceptance.

Please submit your revised manuscript and required files (see below) no later than 21 days from today's (ie 17-Aug-2021) date. Note: the ScholarOne system will 'lock' if submission of the revision is attempted 21 or more days after the deadline. If you do not think you will be able to meet this deadline please contact the editorial office immediately.

on behalf of Prof R. Kerry Rowe (Subject Editor)
openscience@royalsociety.org

Associate Editor Comments to Author:

We've received three referee reports on your paper. Please ensure that you carefully respond to all their comments and concerns while revising your paper (also see the attached reports).

Please submit a point-by-point response detailing what changes you've made upon submission of your revised paper. We'll also require a version of your revised paper which contains either highlighted or tracked changes.

We look forward to receiving your revision!

Reviewer comments to Author:

Reviewer: 1

Comments to the Author(s)

Review Comments: The work represents a 2D study of a PDPF, however, a few points need to be clarified before its consideration for publications.

Comment 1: The main function of a DPF is to filter particles. In this PDPF, most of the airflow and particles will flow through the outlet channel since the inlet channel is blocked. The trap efficiency should be much smaller than the popular DPF. The possible attractive point of this PDPF is its smaller back pressure. However, this is not mentioned in the conclusion. The authors need to provide a comparison of the trap efficiency and back pressure between PDPF and DPF and highlight significances and innovations of the PDPF.

Comment 2: The contents and some figures of this manuscript show high similarity with the reference 20 published in 2009. The authors need to rewrite and restructure most of their contents from Introduction to Section 3.1.

Comment 3: The structure of figure captions is not in general format. The authors need to modify that.

Comment 4: Figure 2 is hard to understand, the authors need to indicate the airflow directions, the parameters mentioned in table 1. Please follow the example from the reference 20 Figure 1(c), that is how a good model structure looks like. Based on the setups of the boundary conditions, the boundary of the inlet channel and the outlet channel (0.04 m ~ 0.294 m) should be wall, not symmetry. The authors need to clarify this.

Comment 5: Line 71, the equations of boundary conditions are unclear, the authors need to clarify what is u_1 and what is u_2 . There is an overuse of brackets. It is suggested to list the boundary conditions of one case in the same bracket. The equation number should be located in the vertical middle.

Comment 6: The figures of the filtration efficiency are lack of the unit. The authors need to clarify if the unit is in % or not. The quality of the Figure 5,6,7 needs to be improved. It is hard to recognize the bar charts, it is suggested to use the line charts for different wall permeabilities.

Comment 7: The authors need to carefully use the signal “-” in table 2. There are only 3 different particle sizes (20 100 1000) and 3 different upstream velocities (1 2 8). There are not enough variables to support the analysis.

Comment 8: The data in Figure 6(d) and Figure 7(d) are missed from 0.8 to 1.0. The authors need to explain why the data are missed.

Comment 9: The fraction efficiencies of different velocities, different particle sizes are almost the same. The impact of an open outlet channel has significant impact on the local velocity field in this PDPF. The authors are suggested to present the velocity field (velocity contours in ANSYS Fluent) for a better understanding.

Comment 10: ANSYS Fluent is a commercial software, the authors need to cite it as required by the ANSYS company.

Reviewer: 2

Comments to the Author(s)

The work concerns the modelling and simulations on the soot laden flow of gas in a partial diesel particulate filter. The research covered an important area and the article may be of interest to many readers. The manuscript has been correctly-structured. Accurate conclusions are presented, which are supported by research results which are original achievement of the Authors. This paper is well written and it is worth to be published in the journal "Royal Society Open Science" with the very minor modifications needed and indicated below:

1. The abstract is well-written, however the main conclusion in the abstract is too long and should be simplified.
2. Necessary references need to be marked, such as the references on Figure 1a and 1b(lines 31-32), and the assumption (2)(lines 31-32).
3. The meaning of each symbol in the formula should be indicated, such as P and μ in the equation (2) ; k_w (permeability ?) in the equation (5) ; Rep in the equation (10) ; dp in the equation (11) ; λ in the equation (12) ; k_B and Δt in the equation (13) , etc.
4. The description of "The continuity equation" in line 70 should be changed to "The continuity equation and momentum equation"
5. Some formulas are marked with incorrect symbol and need to be confirmed by the author. For example: ρ in formula (9) should be ρg ? Re in formula (10) should be Rep ? SCF in line 86 should be italic?
6. Some grammar errors need to be checked throughout, such as "A" in line 54 should be "An"; "has" in line 102 should be "have"; "was" in line 209 should be "is", etc.
7. The url should be added in the "()" in line 241.
8. Is the unit of the flow rate(kg/m³) correct?

Reviewer: 3

Comments to the Author(s)

This is an interesting paper addressing an important issue. It can be considered for publication in Royal Society Open Science. However, I have the following comments that the authors should carefully implement in the revised manuscript before publication.

- 1) Introduction - The sentences "Accordingly, measures are needed to remove the deposited particles periodically. The process is known as the regeneration of DPF." should be substantiated by also citing the following work: Chemical Engineering Science, Volume 137, 2015, Pages 69-78.
- 2) Introduction - The authors wrote "Optimum structure of DPF requires low pressure drop and regeneration frequency.". This is surely true. However, in order to give a more complete picture, they should also highlight in the revised Introduction that, as recently reviewed (see, Catalysts, 2020, 10(11), Article number: 1307), the best strategy to lower both pressure drop and regeneration frequency is to operate a continuous filtration/regeneration process of catalyst-coated DPFs.
- 3) Computational methodology and conditions - Are the computational results grid-independent? The authors should comment on this key issue.
- 4) Results and discussions/Conclusions - In the discussion, the practical impact of the results obtained in this work should be better pointed out. This should also be done in the section "Conclusions".
- 5) Conclusions - The authors should also give an outlook on future research work.

I'm willing to review the revised manuscript.

===PREPARING YOUR MANUSCRIPT===

===PREPARING YOUR REVISION IN SCHOLARONE===

-- If you have uploaded ESM files, please ensure you follow the guidance at <https://royalsociety.org/journals/authors/author-guidelines/#supplementary-material> to include a suitable title and informative caption. An example of appropriate titling and captioning may be found at https://figshare.com/articles/Table_S2_from_Is_there_a_trade-off_between_peak_performance_and_performance_breadth_across_temperatures_for_aerobic_sc_ope_in_teleost_fishes_/3843624.

Author's Response to Decision Letter for (RSOS-211162.R0)

See Appendix A.

RSOS-211162.R1 (Revision)

Review form: Reviewer 1

Is the manuscript scientifically sound in its present form?

Yes

Are the interpretations and conclusions justified by the results?

Yes

Is the language acceptable?

Yes

Do you have any ethical concerns with this paper?

No

Have you any concerns about statistical analyses in this paper?

No

Recommendation?

Accept as is

Comments to the Author(s)

The revised manuscript is great, everything is well structured and clear. I would like to recommend the paper for acceptance on Royal Society Open Science.

Review form: Reviewer 3

Is the manuscript scientifically sound in its present form?

Yes

Are the interpretations and conclusions justified by the results?

Yes

Is the language acceptable?

Yes

Do you have any ethical concerns with this paper?

No

Have you any concerns about statistical analyses in this paper?

No

Recommendation?

Accept with minor revision (please list in comments)

Comments to the Author(s)

The authors have addressed my comments in a satisfactory manner. Overall, the manuscript has been improved after revisions.

I have only one suggestion that could be implemented in the revised manuscript before publication. As ref. [15] is the same as ref. [7], ref. [15] could become: AIChE Journal, Volume 64, Issue 5, 2018, Pages 1714-1722.

Decision letter (RSOS-211162.R1)

Dear Dr Wang

On behalf of the Editors, we are pleased to inform you that your Manuscript RSOS-211162.R1 "2D numerical studies of particle motion and deposition in the channel of diesel particulate filters" has been accepted for publication in Royal Society Open Science subject to minor revision in accordance with the referees' reports. Please find the referees' comments along with any feedback from the Editors below my signature.

Please submit your revised manuscript and required files (see below) no later than 7 days from today's (ie 08-Sep-2021) date. Note: the ScholarOne system will 'lock' if submission of the revision is attempted 7 or more days after the deadline. If you do not think you will be able to meet this deadline please contact the editorial office immediately.

on behalf of Prof R. Kerry Rowe (Subject Editor)
openscience@royalsociety.org

Associate Editor Comments to Author:

The reviewers recommend acceptance, but there is one very minor modification to your bibliography that is recommended. Please ensure this change is made with your revision.

Reviewer comments to Author:

Reviewer: 3

Comments to the Author(s)

The authors have addressed my comments in a satisfactory manner. Overall, the manuscript has been improved after revisions.

I have only one suggestion that could be implemented in the revised manuscript before publication. As ref. [15] is the same as ref. [7], ref. [15] could become: AIChE Journal, Volume 64, Issue 5, 2018, Pages 1714-1722.

Reviewer: 1

Comments to the Author(s)

The revised manuscript is great, everything is well structured and clear. I would like to recommend the paper for acceptance on Royal Society Open Science.

===PREPARING YOUR MANUSCRIPT===

===PREPARING YOUR REVISION IN SCHOLARONE===

Author's Response to Decision Letter for (RSOS-211162.R1)

See Appendix B.

Decision letter (RSOS-211162.R2)

Dear Dr Wang,

I am pleased to inform you that your manuscript entitled "2D numerical studies of particle motion and deposition in the channel of diesel particulate filters" is now accepted for publication in Royal Society Open Science.

on behalf of Prof R. Kerry Rowe (Subject Editor)
openscience@royalsociety.org

Appendix A

Manuscript ID RSOS-211162

Title: 2D numerical studies of particle motion and deposition in the channel of diesel particulate filters

Authors: Xiaolong Wang, Yangbo Deng, Yang Liu

Article Type: Original Research Paper

Dear Editor and Reviewers,

We appreciate very much the helpful comments by the reviewers and the editor. The English is also carefully checked and polished. Following is the responses to all comments point by point. We numbered the comments and gave answers. All the revised parts or added content are distinguished by yellow base. Our responses on their questions and suggestions are as follows.

Editor comments:

1. We've received three referee reports on your paper. Please ensure that you carefully respond to all their comments and concerns while revising your paper (also see the attached reports).

ANSWER: We accept this comment. Following is the responses to all comments point by point.

Reviewer: 1

Review Comments: The work represents a 2D study of a PDPF, however, a few points need to be clarified before its consideration for publications.

1. The main function of a DPF is to filter particles. In this PDPF, most of the airflow and particles will flow through the outlet channel since the inlet channel is blocked. The trap efficiency should be much smaller than the popular DPF. The possible attractive point of this PDPF is its smaller back pressure. However, this is not mentioned in the conclusion. The authors need to provide a comparison of the trap efficiency and back pressure between PDPF and DPF and highlight significances and innovations of the PDPF.

ANSWER: We accept this comment. In the original manuscript, by the utilization of the definition of pressure drop reduction efficiency η_P , a comparison of the trap efficiency and back pressure between PDPF and DPF has been presented in the original Figure 8. Please see the re-numbered Figure 9 and the commented part. However, this issue was ignored in the conclusion part. In the revised manuscript, we have commented on this issue. Please see line 225, page 13. “The main function of a DPF is to filter particles. In the PDPF, most of the airflow and particles will flow through the outlet channel since the inlet channel is blocked, which is particularly notable at low wall permeabilities, resulting in the trap efficiency being smaller than that of the popular DPF. However, the attractive point of this PDPF is its smaller back pressure. This contradictory relationship between the trap efficiency and the back pressure can be harmonized by its wall permeability.”

2. The contents and some figures of this manuscript show high similarity with the reference 20 published in 2009. The authors need to rewrite and restructure most of their contents from Introduction to Section 3.1.

ANSWER: We accept this comment. Indeed, the literature has inspired us a lot to write this paper. For the journal's submission requirements, we checked the duplicates in advance, and the papers met the requirements in terms of repetition rate. According to reviewer's, the contents from Introduction to Section 3.1 have been rewritten and restructured in the revised manuscript, please see the revised part.

3. The structure of figure captions is not in general format. The authors need to modify that.

ANSWER: We accept this comment, please see the revised figure captions.

4. Figure 2 is hard to understand, the authors need to indicate the airflow directions, the parameters mentioned in table 1. Please follow the example from the reference 20 Figure 1(c), that is how a good model structure looks like. Based on the setups of the boundary conditions, the boundary of the inlet channel and the outlet channel (0.04 m -0.294 m) should be wall, not symmetry. The authors need to clarify this.

ANSWER: We accept this comment. We have redrawn the model structure by following the example from the reference 20. We have re-checked the setups of the boundary conditions and found that the boundary condition of the inlet channel as well as the outlet channel (0.04 m - 0.294 m) is symmetry as used in reference 20.

5. Line 71, the equations of boundary conditions are unclear, the authors need to clarify what is u_1 and what is u_2 . There is an overuse of brackets. It is suggested to list the boundary conditions of one case in the same bracket. The equation number should be located in the vertical middle.

ANSWER: We accept this comment. We have re-organized the contents of boundary condition, please see the revised comments of boundary conditions.

6. The figures of the filtration efficiency are lack of the unit. The authors need to clarify if the unit is in % or not. The quality of the Figure 5,6,7 needs to be improved. It is hard to recognize the bar charts, it is suggested to use the line charts for different wall permeabilities.

ANSWER: We accept this comment, this is a good suggestion. We re-checked the unit of filtration efficiency and found it was correct. It is true that the quality of the original Figure 5, 6, 7 need to be improved. The re-numbered Figures 5, 7, 8 were revised and we used the line charts for different wall permeabilities, please see the revised Figures.

7. The authors need to carefully use the signal “-“ in table 2. There are only 3 different particle sizes (20 100 1000) and 3 different upstream velocities (1 2 8). There are no enough variables to support the analysis.

ANSWER: This is a very good suggestion. We are sorry for the mistake, we have corrected the errors, please see the revised Table. 2.

8. The data in Figure 6(d) and Figure 7(d) are missed from 0.8 to 1.0. The authors need to explain why the data are missed.

ANSWER: We accept this comment. The original Figure 6 (d) and Figure 7 (d) show the particle deposition distribution under a low wall permeability, where little airflow and few particles flow into the inlet channel. We have explained on this issue in

section 3.3, “It can be seen obviously that most particles are deposited in rear of the channel at a high permeability and deposited towards the front of inlet channel as permeability decreases. It meets our common knowledge that a lower permeability leads to an increased resistance of gas flow in the inlet channel, and then shortens the moving distance of particles. Subject to large flow resistance in these cases , particles are hard to reach the end of the inlet channel [12]”.

9. The fraction efficiencies of different velocities, different particle sizes are almost the same. The impact of an open outlet channel has significant impact on the local velocity field in this PDPF. The authors are suggested to present the velocity field (velocity contours in ANSYS Fluent) for a better understanding.

ANSWER: We accept this comment. There are three operation conditions that may affect deposition distribution of particles in a single-channel PDPF studied in our paper, among which the wall permeability has the most primary effect. The dependence of the particle motion trajectory on upstream velocities and particle sizes is significant at a high wall permeability. However, this effect is alleviated in the cases of small permeabilities, leading to the difference in the fraction efficiencies of different velocities, different particle sizes being not remarkable. Indeed, the velocity field in a PDPF is very different from that in a regular DPF, which has been reported in detail in reference 14. Similar phenomena have also been observed in our numerical simulations. For a better understanding, we have presented the velocity field of continuous phase under two different wall permeabilities (the computational

domain was scaled along the transverse direction using a scaling factor equal to 8).

Please see Figure 6.

10. ANSYS Fluent is a commercial software, the authors need to cite it as required by the ANSYS company.

ANSWER: We accept this comment. We cite the software, please see reference [19].

[19] ANSYS Fluent Theory Guide, Release 17.0, ANSYS Inc., Canonsburg, 2016.

Reviewer: 2

Comments to the Author(s)

The work concerns the modelling and simulations on the soot laden flow of gas in a partial diesel particulate filter. The research covered an important area and the article may be of interest to many readers. The manuscript has been correctly-structured. Accurate conclusions are presented, which are supported by research results which are original achievement of the Authors. This paper is well written and it is worth to be published in the journal “Royal Society Open Science” with the very minor modifications needed and indicated below:

1. The abstract is well-written, however the main conclusion in the abstract is too long and should be simplified.

ANSWER: We accept this comment. The main conclusion in the abstract has been simplified. Please see the revised part.

2. Necessary references need to be marked, such as the references on Figure 1a and 1b(lines 31-32), and the assumption (2)(lines 31-32).

ANSWER: We accept this comment. We have cited reference mentioned by the reviewer, please see the revised Figure 1a and line 55, page 3.

3. The meaning of each symbol in the formula should be indicated, such as

P and μ in the equation (2) ; k_w (permeability?) in the equation (5) ; Re_p in the equation (10) ; dp in the equation (11) ; λ in the equation (12) ; k_B and Δt in the equation (13) , etc.

ANSWER: We accept this comment. The meaning of each symbol in the formula was expressed, please see the revised formula (2), (5), (10)-(13).

4.The description of “The continuity equation” in line 70 should be changed to “The continuity equation and momentum equation”

ANSWER: We accept this comment. “The continuity equation” has been replaced as “The continuity equation and momentum equation”, please see line 66, page 4.

5.Some formulas are marked with incorrect symbol and need to be confirmed by the author. For example: ρ in formula (9) should be ρ_g ? Re in formula (10) should be Re_p ? SCF in line 86 should be italic?

ANSWER: We accept this comment. Yes, ρ in revised formula (5) should be ρ_g , Re in formula (6) should be Re_p , SCF in line 86 should be italic, please see the revised formula (5), formula (6) and SCF.

6. Some grammar errors need to be checked throughout, such as “A” in line 54 should be “An”; “has” in line 102 should be “have”; “was” in line 209 should be “is”, etc.

ANSWER: We are very sorry for the mistake. The grammar errors pointed out by the reviewer have been corrected. At the same time, grammar errors have been checked and corrected throughout the manuscript.

7. The url should be added in the “()” in line 241.

ANSWER: We are sorry for the mistake. The url has been added in the (), please see revised reference [9].

8. Is the unit of the flow rate (kg/m^3) correct?

ANSWER: We are sorry for the mistake. The unit of the flow rate should be kg/s , please see line 121, page 7.

Reviewer: 3

This is an interesting paper addressing an important issue. It can be considered for publication in Royal Society Open Science. However, I have the following comments that the authors should carefully implement in the revised manuscript before publication.

1. Introduction - The sentences “Accordingly, measures are needed to remove the deposited particles periodically. The process is known as the regeneration of DPF.” should be substantiated by also citing the following work: Chemical Engineering Science, Volume 137, 2015, Pages 69-78.

Sarli V D, Benedetto A D. 2015 Modeling and simulation of soot combustion dynamics in a catalytic diesel particulate filter, *Chemical Engineering Science*, **137(1)**, 69-78. (10.1016/j.ces.2015.06.011)

ANSWER: We accept this comment. The reference mentioned by the reviewer was cited in the revised manuscript, please see line 35, page 2.

2. Introduction - The authors wrote “Optimum structure of DPF requires low pressure drop and regeneration frequency.”. This is surely true. However, in order to give a more complete picture, they should also highlight in the revised Introduction that, as recently reviewed (see, *Catalysts*, 2020, 10(11), Article number: 1307), the best strategy to lower both pressure drop and regeneration frequency is to operate a continuous filtration/regeneration process of catalyst-coated DPFs.

ANSWER: We agree with these comments. We cited the reference suggested by the reviewer and added comment on this issue, “the best strategy to lower both pressure drop and regeneration frequency is to operate a continuous filtration/regeneration process of catalyst-coated DPFs.”, please see line 39, page 2.

Lisi L, Landi G, Sarli V D. 2020 The issue of soot-catalyst contact in regeneration of catalytic diesel particulate filters: a critical review. *Catalysts*, **10(11)**, 1307. (10.3390/catal10111307)

3. Computational methodology and conditions - Are the computational results grid-independent? The authors should comment on this key issue.

ANSWER: We accept this comment. We have commented on this issue, “the grid independence of the results is verified.”, please see line 95, page 6.

4. Results and discussions/Conclusions – In the discussion, the practical impact of the results obtained in this work should be better pointed out. This should also be done in the section "Conclusions".

ANSWER: We accept this comment. Both in the discussions and the section “Conclusions”, We have commented on the practical impact of the results obtained in this work. Please see line 210-212, page 12 and line 225-227, page 13.

5. Conclusions - The authors should also give an outlook on future research work.

ANSWER: We accept this comment. We have commented on this issue in the section “Conclusions”, please see line 233, page 13. “Under the conditions investigated in this work, with the progress of particle deposition, the wall permeability decreases, which will be bound to reduce the trap efficiency of PDPF. Accordingly, regeneration procedure must be started when the particle deposition reaches a certain level. One promising way to prolong the working cycle of PDPF may be to optimize its structure. Future work will attempt to investigate the performance of PDPF with asymmetric channels.”

Sincerely yours

Xiaolong Wang and Yangbo Deng

Appendix B

Manuscript ID RSOS-211162.R2

Title: 2D numerical studies of particle motion and deposition in the channel of diesel particulate filters

Authors: Xiaolong Wang, Yangbo Deng, Yang Liu

Article Type: Original Research Paper

Dear Editor and Reviewers,

We appreciate very much the helpful comments by the reviewers and the editor. The English is also carefully checked and polished. Following is the responses to all comments point by point. We numbered the comments and gave answers. All the revised parts or added content are distinguished by yellow base. Our responses on their questions and suggestions are as follows.

Associate Editor Comments to Author:

1. At Step 3 'File upload' of the ScholarOne electronic submission form, you should now include the following files:

ANSWER: We accept this comment. An individual file of each figure has been uploaded.

ANSWER: We accept this comment. An editable file of each table has been uploaded.

ANSWER: We accept this comment. An editable file of all figure and table captions has been uploaded. We did not have ESM to upload.

ANSWER: We did not need to provide image files for potential cover images.

5. -- A copy of your point-by-point response to referees and Editors.

ANSWER: We accept this comment. A copy of your point-by-point response to referees and Editors has been uploaded.

6. Please note that, if you are requesting a discretionary waiver for the article processing charge, the waiver form must be included at this step. Please see

<https://royalsocietypublishing.org/rsos/waivers#question4>. Please add these files and then resubmit the paper for consideration.

ANSWER: We did not need requesting a discretionary waiver for the article processing charge.

Reviewer: 3

Comments to the Author(s)

The authors have addressed my comments in a satisfactory manner. Overall, the manuscript has been improved after revisions.

I have only one suggestion that could be implemented in the revised manuscript before publication. As ref. [15] is the same as ref. [7], ref. [15] could become: AICHE Journal, Volume 64, Issue 5, 2018, Pages 1714-1722.

ANSWER: We accept this comment. The ref. [15] has been replaced with the suggested reference.

Sincerely yours

Xiaolong Wang and Yangbo Deng